# Triple Jeopardy of Minority Status, Social Stressors, and Health Disparities on Academic Performance of College Students

**DOI:** 10.3390/ijerph20136243

**Published:** 2023-06-28

**Authors:** Monideepa B. Becerra, Rushil J. Gumasana, Jasmine A. Mitchell, Saba Sami, Jeffrey Bao Truong, Benjamin J. Becerra

**Affiliations:** Center for Health Equity, Department of Health Science and Human Ecology, California State University-San Bernardino, 5500 University Parkway, San Bernardino, CA 92407, USA; rushil.gumasana@csusb.edu (R.J.G.); 007438243@coyote.csusb.edu (J.A.M.); 006245222@coyote.csusb.edu (S.S.);

**Keywords:** mental health, social stressors, academic performance, pandemic, COVID-19, young adults, college students

## Abstract

**Objective:** In this study, we evaluated the role of minority status, as well as pandemic-related social stressors and health disparities on short- and long-term academic performances of college students. **Methods:** Cross-sectional analysis using descriptive and bivariate statistics were used to identify participants of minority status as well as pandemic-related prevalence of social stressors and health disparities and their roles on academic performance. **Results:** Poor academic performance was significantly related to being food insecure, experiences of discrimination, serious psychological distress, and low daytime wakefulness during the pandemic but only significantly based on minority status. Grade point average was significantly associated with serious psychological problems among males, independent of race/ethnicity identity. **Conclusions:** Institutes of higher education, when tasked with developing post-pandemic policies to address equity gaps in academic success, may benefit their students by integrating system-wide holistic approach to support, including interventions on basic needs support and health and resilience building.

## 1. Introduction

The COVID-19 pandemic, attributed to over 960,000 deaths in the U.S. [1] and a projected economic cost of $16 trillion [2], remains an ongoing public health threat that has impacted nearly every aspect of daily life, including healthcare utilization among at-risk groups [3], joblessness [4], mental health [5], etc. Furthermore, due to the burden of Long COVID, which includes ongoing signs and symptoms, the burden of the novel disease is not truly alleviated [6]. A key area of impact has also been among college students, due to the abrupt shift from primarily face-to-face learning to virtual [7]. Empirical evidence on addressing this global shift among students at all levels highlights that such a change in learning has been coupled with a plethora of barriers, including technological limitations [8,9,10], lack of faculty–student interactions and communications [11,12], lack of motivation [11,13], and increased workload [10,11].

For example, a study among Georgia State Public Health students found that concerns over being infected, having a family member be infected with the COVID-19 virus, and increases in workload in the form of new assignments were prevalent among the population surveyed [10]. Likewise, an assessment among Public Health Nutrition students further highlighted that technological issues at home, reduced motivation and effort, as well lack of interaction with fellow students were common barriers cited [13].

However, global assessments on the impact of the COVID-19 pandemic on academic performance, although limited, have provided inconsistent outcomes. For example, in a study assessing the impact of COVID-19 on academic performance of students, Engelhardta et al. not only reported that there were no differences in test results during the pandemic to data from pre-pandemic, the equity gap among females students actually narrowed [14]. Likewise, in a study with over 370 college students in Egypt, Said et al. found no significant differences before or during the pandemic [15]. On the other hand, in Operative Dentistry preclinical courses at Harvard, Inoue et al. found that student performance during the pandemic improved when compared to pre-pandemic outcomes [16]. Similarly, Rodríguez-Planas noted that low-income college students in the lower quartile of the grade point average actually outperformed their high-income counterparts, which was further attributable to flexible grading options [17].

Such inconsistencies may be attributable to various factors, including insight into a specific discipline, lack of homogeneity in measuring academic performance measures, diversity of studies in terms of country and population, as well as lack of assessment of societal factors that may lead to a cumulative burden on academic performance among the most at-risk groups. As such, while such literature provides the importance of addressing how academic performance has been impacted by COVID-19, the body of evidence remains limited and does not provide insight among the most vulnerable. In this study, we aimed to address the role of the pandemic, associated changes in social stressors and health disparities, and the associated burden on college students’ academic performance, delineated by minority status. Our study is thus driven by the theoretical underpinnings of intersectionality: a framework that acknowledges how societal categorizations, such as race, ethnicity, gender, etc., can create overlapping experiences that lead to disparities in outcomes [18,19]. Our goal is to expand the current use of intersectionality and address the role of existing health disparities and how they may further be interdependent with social factors to create additional disproportionate outcomes among the most vulnerable. As such, this study aims to understand how the pandemic has impacted college students’ academic performance, considering their unique societal experiences and identities as key to ensuring targeted prevention strategies and support measures for those at most need, and it is hypothesized that those experiencing multiple stressors and disparities would likely have worsened outcomes.

## 2. Methods

Data were collected from a mid-sized four-year public university, which was also a federally designated Hispanic serving and minority serving institution in the United States. A majority of the students were racial/ethnic minorities (>75%), first-generation (81%), and of low socioeconomic status (58%). All students enrolled on campus aged 18 years or older were eligible to participate. Courses with a diversity of majors were selected, and instructors were asked to disseminate the survey, with the option of extra credit.

Students were given an online anonymous survey of a total of 35 questions, some with multiple components, which assessed their grade level, minority status (sex at birth and race/ethnicity), social stressors (food insecurity, experiences of discrimination), health (sleep health, mental health), as well as academic performance (GPA, grades during pandemic) as part of an annual needs assessment. Questions relevant to the scope of this study were included for assessment.

We did not have any respondent report other or intersex for sex at birth, and thus the variable was dichotomized as male, female. Race/ethnicity was assessed as Hispanic/Latino, non-Hispanic African American/Black, non-Hispanic Asian/Pacific Islander, non-Hispanic white, two or more, or other, as is routine with national and state surveys of U.S. race/ethnicity identification, as well as how race/ethnicity was identified at the institution, thus making our data comparable. Hispanic/Latino and non-Hispanic African American/Black and non-Hispanic Asian/Pacific Islander were collapsed as racial/ethnic minority due to majority of the study population being Hispanic/Latino, similar the institution’s demographic makeup. We did not have any students reporting two or more or other for race/ethnicity. Grade level was further assessed as first/second year, third/fourth year, fifth year or more, and graduate student.

The social stressor of food insecurity, which is a marker for low socio-economic status, was measured using the United States Department of Agriculture [20] questionnaire, experiences of discrimination were assessed using the everyday discrimination scale (EDS) [21], and, consistent with a previous assessment [22], mean EDS score ranged from 0 to eight, with higher scores indicative of more experiences of discrimination.

Mental health was assessed using the Kessler-6 scale for serious psychological distress, which has been validated in the literature [23]. A score of 8–12 was considered mild-to-moderate distress, whereas 13 or higher was categorized as serious psychological distress (SPD) [24,25]. Sleep health was assessed using daytime wakefulness, where participants were asked if they felt tiredness/fatigue/sleepiness during daytime, from the Berlin sleep questionnaire [26,27] and has been previously used among college students [22].

Academic performance was measured using GPA as well as participants’ reporting whether their grade worsened during the pandemic. We aimed to include students’ perception, as GPA requires several units over a period of time [28] to see substantial change, and short-term interferences of one–two semesters may not drastically change GPA.

To assess the prevalence of study population characteristics and our variables of interest, descriptive statistics were used. To evaluate associations between academic performance and each of our variables of interest, bi-variate statistics (chi-square, Fisher’s exact, independent sample *t*-tests) were conducted. All data were further sub-grouped by sex and race/ethnicity in order to identity the most at-risk groups. Missing data were excluded from all statistical analyses. All data were analyzed in SPSS version 28 (IBM, Corp.), and *p* values of less than 0.05 were used to denote significance.

## 3. Results


**Study population characteristics**


Table 1 displays the study population characteristics with a majority being third- or fourth-year students (40.4%), females (73.1%), and racial/ethnic minorities (81.3%). Among social stressors, 38.8% reported being food insecure, and a mean EDS score of 2.44 was noted among the study population as well. Among measures of health disparities, 83% reported low daytime wakefulness, whereas 22% had SPD. For our academic performance variables of interest, the average GPA was 3.2, whereas 47% reported that the pandemic worsened their grades.


**Academic performance and minority status**


Table 2 illustrates the association between each minority status variable and academic performance. A significantly higher percent of females in the overall population, compared to males, reported that their grades had worsened due to the pandemic (52.0% vs. 35.7%, *p* < 0.05). Multivariable binary logistic regression analysis demonstrated that females reported significantly higher odds of worsening grades due to the pandemic (odds ratio [OR = 1.95, 95% confidence interval [CI]: 1.04–3.67; *p* < 0.05) when compared to males. When evaluating GPA, racial/ethnic minority students had significantly lower GPA compared to their counterparts (3.23 vs. 3.44, *p* <.05), although sex and GPA did not yield any significant association. Multivariable linear regression analysis notef that minority status was associated with 0.206 units less GPA on average (*p* < 0.05).


**Academic performance and social stressors**


As shown in Table 3, the prevalence of worsened grades due to the pandemic was found among food-insecure participants, compared to their food secure counterparts (60.5% vs. 38.6%, *p* < 0.05). Sub-group analyses showed that 42.1% of females compared to 29.6% of males, and 41.3% of minority students compared to 28.2% of non-minority students, reported food insecurity. Furthermore, our results showed that food-insecure racial/ethnic minority female students reported a higher prevalence of reporting that the pandemic had a negative impact on grades when compared to those with similar demographics but who were food secure (64.8% vs. 43.5%, *p* < 0.05). GPA was not significantly related to food insecurity in the population. We further conducted multivariable binary logistic regression analyses to assess the role of food insecurity on perceived worsening of grades during the pandemic after adjusting demographic characteristics. The results noted that food-insecure participants were 138% more likely to report worsening grades during the pandemic than their food-secure counterparts (OR = 2.38, 95% CI: 1.33–4.27).

When evaluating the relationship between discrimination and academic performance, we found that those who had worsened grades during the pandemic also reported a significantly higher mean EDS (mean, standard error (SE): 3.24, 0.24), as compared to their counterparts (mean, SE: 1.74, 0.17). Sub-group analyses demonstrated that the association between EDS and worsened grades during the pandemic remained significant for racial/ethnic minority females (1.88 vs. 2.95, *p* < 0.05), racial/ethnic minority males (1.70 vs. 4.69, *p* < 0.05), and non-racial/ethnic minority females (3.43 vs. 1.40, *p* < 0.05) but not for non-racial/ethnic minority males (2.00 vs. 1.75, *p* ≥ 0.05). Multivariable binary logistic regression analysis further highlighted that the participants had a 48% increase in the odds of reporting worsening grades during the pandemic (OR = 1.48, 95% CI: 1.23–1.64, *p* < 0.001) for every increase in EDS score. EDS and GPA were not significantly associated in the full population or in sub-group analyses utilizing either bivariate or regression analyses.


**Academic performance and health disparities**


As shown in Table 4, participants with SPD had a significantly lower mean GPA as compared to those without SPD (3.10 vs. 3.33, *p* < 0.05). In addition, a significantly higher prevalence of worsened grades during the pandemic were reported among those with SPD (76.6% vs. 39.9%, *p* < 0.05) as compared to their counterparts.

We further evaluated if the association between mental health disparities and academic performance persisted by sex and race/ethnicity to further elucidate the most vulnerable. The association between SPD and GPA only remained significant for racial/ethnic minority males (2.62 vs. 3.38, *p* < 0.05) but no other subgroups.

Racial/ethnic minority females with SPD (vs. no SPD) also had significantly higher prevalence of worsened grades during the pandemic (77.8% vs. 45.8%, *p* < 0.05). This relation persisted for racial/ethnic males as well (84.6% vs. 15.2%, *p* < 0.05) but not males or females of a non-racial/ethnic minority identity.

As further shown in Table 4, a higher prevalence of worsened grades during the pandemic were noted among those who also reported low daytime wakefulness (52.9% vs. 20.0%, *p* < 0.05) compared to those who did not report such poor sleep outcomes.

Multivariable binary logistic regression models adjusted for demographic characteristics also noted that participants who had SPD were five-and-a-half times more likely to report worsening grades during the pandemic than their counterparts with no SPD (OR = 5.54, 95% CI: 2.57–11.92, *p* < 0.001). In addition, the results of multivariable linear regression analysis showed that participants with SPD had 0.222 less GPA on average than those without SPD (*p* < 0.05). Likewise, multivariable binary logistic regression models adjusted for such characteristics also noted that participants who reported low daytime wakefulness were almost four times more likely to report worsening grades during the pandemic than their counterparts with no SPD (OR = 3.97, 95% CI: 1.63–9.69, *p* < 0.01). Consistent with the bivariate results, regression analysis did not demonstrate an association between daytime wakefulness and mean GPA.

Sub-group analyses, not presented in table, further showed that racial/ethnic minority females who reported low daytime wakefulness (vs. not) had a higher prevalence of worsened grades during the pandemic (56.5% vs. 26.7%, *p* < 0.05), and such an association was also found for racial/ethnic males as well (45.5% vs. 7.7%, *p* < 0.05) but not for non-racial/ethnic minority females or males. GPA did not yield significance in any analyses.

Cumulatively, as shown in Table 5, worsened grades during the pandemic were significantly associated with food insecurity, experiences of discrimination, SPD, and low daytime wakefulness among racial/ethnic minority females. Among racial/ethnic minority males, worsened grades were associated with experiences of discrimination, SPD, and low daytime wakefulness. Among non-racial/ethnic females, only experiences of discrimination were associated with worsened grades during the pandemic. A long-term measure of academic performance, GPA, was significantly associated with SPD among males, independent of race/ethnicity identity.

## 4. Discussion

We aimed to evaluate the intersectionality of minority status, as well as social stressors and health disparities during the COVID-19 pandemic, on academic performance of college students. Our results highlighted that racial/ethnic minority female college students experienced the most burden, with worsened academic performance during the pandemic related with social stressors (food insecurity, experiences of discrimination), as well as health disparities (psychological distress, low daytime wakefulness).

The literature on the minority stress model notes that members of marginalized groups who often experience an accumulation of chronic stressors may have worsened disparities [29,30], which, in turn, highlights the importance of addressing the intersectional burden of such experiences on health and related outcomes. As such, our results may have been indicative of heightened allostatic burden among the most vulnerable from the intersectional burden of such minority stressors. Additionally, the Institute of Medicine’s landmark report [31] noted that the disproportionate burden among racial/ethnic minority women remains a public health concern despite the interventions to reduce cardiovascular disease-related disparities among women. Likewise, delays in breast cancer screening [32], potentially attributable to inadequate access to care among minority women [33], pregnancy-related deaths [34], etc., have been shown to disproportionately impact women of racial/ethnic minority status. Cumulatively, while such empirical evidence highlights the increased disparities noted among racial/ethnic minority women, our study results further expand the body of knowledge by providing insight into the increased academic disparities experienced in the population.

Furthermore, the literature highlights substantial prevalence of experiences of discrimination among women, especially those of an additional minority status [35]. Likewise, differences in psychological distress between men and women have also been attributable to increased experiences of sexism among women as well [36]. This may further explain our results, where female college students who reported experiences of discrimination also had worsened academic performance and thus public health- and campus-based initiatives on resilience building, social networks, self-efficacy, stress inoculation, etc. [37], against such experiences may need targeted approaches among subgroups.

We also found that nearly 39% of our study participants were food insecure, as compared to 10.5% of the U.S. adult population [38] and 29% of U.S. four-year college students [39] during the pandemic, and a higher prevalence of food insecurity was also found among females and racial/ethnic minorities in our study. The literature demonstrates that food insecurity is substantially associated with poor health outcomes, including increased weight among women [40,41], as well as overall poor health status among college students [42]. In a recent study on assessing food insecurity among college students, Owen et al. [43] noted that approximately 34% of the study participants were food insecure. While our results showed a comparable prevalence, we further highlighted the heightened burden among minority students (race/ethnicity and sex) and the impact on worsened grades during the pandemic. Likewise, additional studies have found that that food-insecure college students were more likely to withdraw from courses [44] or have lower GPA scores [45]. While GPA is a marker for long-term academic success, our results, demonstrating the immediate short-term academic outcomes (worsened grades) associated with past-year food insecurity coupled with the literature, highlighted both the short- and long-term burden of such a social stressor on academic outcomes of the most vulnerable. However, while not all campus-based food pantries have been successful, providing such resources along with a de-stigmatization of food insecurity [46], they may provide a scope of needed intervention.

In addition, when evaluating health disparities and academic performances, our study results showed that low daytime wakefulness and SPD were associated with worsened grades among both racial/ethnic minority males and females. The literature notes that minority students may often experience lack of inclusiveness [47], face barriers to accessing campus resources [48], have additional family responsibilities [49], need to manage work–school balance to meet financial responsibilities of higher education [50], etc., and thus such additional factors may contribute to less-than-optimal academic outcomes. As such, campus-based initiatives that mitigate such burdens are of critical need.

Nevertheless, it should be noted that SPD, a marker for serious mental illness [23], was also associated with the long-term assessment of academic performance GPA among males, independent of race/ethnicity, but not our short-term academic performance measure of interest. Therefore, such a negative long-term academic outcome related to SPD is likely not a result of the pandemic alone and may be further attributable to a more severe accumulation of stressors coupled with social stigma related to mental health among men.

The results of this study should be evaluated in the context of limitations inherent to cross-sectional design. Such a study design did not allow for causal or temporal relationship assessments. Furthermore, the survey design was susceptible to recall bias. Our study population was also primarily Hispanic, and thus future studies of survey questions and methodologies need to be implemented. Such limitations notwithstanding, our results show the need for targeted interventions to optimize academic success among racial/ethnic minorities and the imperative need to address the mechanisms and interventions that address the interplay between minority status, societal stressors, and health disparities. For example, to mitigate the burden of equity gaps in academic performance that are often reported across institutes of higher education [51,52], a paradigm shift is of imperative need to move from traditional individual-level only approaches, such as tutoring, loans, etc., to that of a holistic approach that promotes system-level interventions to alleviate the burden of social stressors and emergent health disparities. Based on the results of our study, such interventions may include routine screening for mental illness, especially in high enrollment classes that provide larger access to students [53], the availability of food pantries coupled with campus-wide initiatives to reduce social stigma related to food pantry use [46], the inclusion of nap areas on campus [54], sleep literacy interventions [55,56], etc., and assessments of how such interventions effectively reduce the equity gaps in short- and long-term academic performances among the most vulnerable students are needed. It should also be noted that Long COVID and its associated impact remains while the pandemic’s worse era has receded, with the literature highlighting the substantial burden to physiological outcomes [57]. Research remains on how that further impacts other outcomes, including mental health, social stressors, as well as academic performance.

## 5. Conclusions

Our results provided a unique insight into the burden of the pandemic among college students’ academic performances. While the literature remains inconsistent on such an association, our study evaluated worsened societal factors and health disparities during the COVID-19 pandemic and their association with academic performances, thus providing scopes of interventions during public health emergencies. As educational systems are faced with the need to implement post-pandemic initiatives, our results call to action that a system-wide holistic approach to student support (i.e., interventions on sleep literacy, system-wide routine efforts to provide and/or refer students to resources on and off campus on food needs, mental health services that target gender-based stigma, efforts to mitigate discrimination, as well as empowering students with coping mechanisms related to such experiences) becomes an integral component of the “new normal” in post-pandemic education.

## Figures and Tables

**Table 1 ijerph-20-06243-t001:** Study population characteristics, *n* = 223.

Academic Grade Level	
First or second, %	26.9
Third or fourth, %	40.4
Fifth or more, %	32.7
Minority status	
Females, %	73.1
Racial/ethnic minority, %	81.3
Social stressors	
Food insecure, %	38.8
Mean everyday discrimination score (standard deviation)	2.44 (2.20)
Health disparities	
Low daytime wakefulness, %	83.3
Serious psychological distress, %	22.4
Academic performance	
Mean GPA (standard deviation)	3.27 (0.49)
Pandemic worsened grades, %	47.1

**Table 2 ijerph-20-06243-t002:** Association between minority status and academic performance. * denotes *p* < 0.05.

	% Reporting Pandemic-Worsened Grades	Mean Grade Point Average
Sex	*	
Females	52.0	3.29
Males	35.7	3.24
Racial/ethnic group		*
Racial/ethnic minority	47.9	3.23
Not racial/ethnic minority	46.2	3.44

**Table 3 ijerph-20-06243-t003:** Association between food insecurity and academic performance. * denotes *p* < 0.05.

	% Reporting Pandemic-Worsened Grades
Food security status, full population	*
Food insecure	60.5%
Food secure	38.6%
Food security status, racial/ethnic minority, female	*
Food insecure	64.8%
Food secure	43.5%
Food security status, racial/ethnic minority, male	
Food insecure	53.3%
Food secure	24.1%
Food security status, non-racial racial/ethnic minority, female	
Food insecure	60.0%
Food secure	42.1%
Food security status, non-racial/ethnic minority, male	
Food insecure	0.0%
Food secure	44.4%

**Table 4 ijerph-20-06243-t004:** Association between health disparities and academic performance. * denotes *p* < 0.05.

	% Reporting Pandemic-Worsened Grades	Mean Grade Point Average
Serious psychological distress (SPD)	*	*
SPD	76.6	3.33
Not SPD	38.9	3.10
Daytime wakefulness	*	
Low	52.9	3.26
Not low	20.0	3.34

**Table 5 ijerph-20-06243-t005:** Factors associated with low academic performance among college studies, sub-grouped by minority status.

	Food Insecurity	Discrimination	Serious Psychological Distress	Low Daytime Wakefulness
Racial/ethnic minority, female	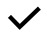 1	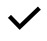 1	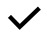 1	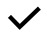 1
Racial/ethnic minority, male		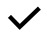 1	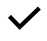 1, 2	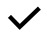 1
Non-racial/ethnic minority, female		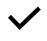 1		
Non-racial/ethnic minority, male			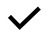 2	

1 = associated with reporting lower grades during the pandemic; 2 = associated with lower mean GPA.

## Data Availability

Data is not available to share per approval standards.

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
