# Peer review of "Triple Jeopardy of Minority Status, Social Stressors, and Health Disparities on Academic Performance of College Students"

_ijerph, 2023, doi:10.3390/ijerph20136243_

Round 1

Reviewer 1 Report

The authors first identified inconsistencies between literatures about COVID's impact on student's academic performance, and hypothesized the cause from various factors.

To reduce the impact from limitations, authors decided to restrict study subjects to the most vulnerable group. It is a very reasonable strategy, please also discuss the limitations of such decision on the paper's conclusions: it is limited to the specific timepoint after pandemic, and among a specific college's students. The generalisability will be limited and shall be addressed and discussed. Author then selected the target population that represents the study population of interest. All questionnaires developed to address study questions are also clearly stated. The aspects all make perfect sense. The study's results are very informative and indicative to identify prevalence of issues from various dimensions among minority colleague students. Identifying the issue is a critical first step to actions. Such methods can be replicated in other groups to assess severity across demographics. Below are some suggestions from a statistician's perspective:

1. there are multiple dimensions, such as social stressor, food insecurity, mental and sleep health. It would be helpful to come up with a score using tools such as PCA or structural equation modeling, or at least a multivariable regression to understand each component's contributions.

2. COVID is a factor affecting all factors: social stressors, health disparities, food insecurity, etc. Therefore it maybe a multiple layer structure as assumed in the study: COVID->factors->academic performance. On the other hand, academic performance may just another direct consequence from COVID, irrelevant from the assumed factors. There can be causal model attempts to understand the causal pathways.

3. That's largely in the Hispanic group, may need consistence check among other minority groups.

4. Student's perception is a relatively subjective measure out outcome, compared to GPA, maybe a good idea to repeat the study measurement later. The perception has strong differences between in various comparisons while GPA doesn't show significance among some, which is a reflection of such limitation. Also because student's perception is a relatively new measurement, there's no rules to identify a "scientifically meaningful" change. It may be validated later with more studies.

5. A suggestion: stratified has specified statistical meanings. Please try to use "subgroup analysis" if it better describes the analysis performed.

6. Suggestion: add the size of each category into the table, which will make the table more informative, such as table 3.

7. There's no multiplicity strategy, so the overall type I error rate is not controlled. Also we wish to understand the magnitude of effect, a measurement such as odds ratio, risk difference would help.  

Overall, it is a interesting study investigating a critical topic.

Thank you for your work!

Author Response

Dear reviewer,

Thank you for your feedback. We have made the necessary and feasible changes and added the responses in bold for ease of finding.

Reviewer 1

The authors first identified inconsistencies between literatures about COVID's impact on student's academic performance, and hypothesized the cause from various factors.

To reduce the impact from limitations, authors decided to restrict study subjects to the most vulnerable group. It is a very reasonable strategy, please also discuss the limitations of such decision on the paper's conclusions: it is limited to the specific timepoint after pandemic, and among a specific college's students. The generalisability will be limited and shall be addressed and discussed. Author then selected the target population that represents the study population of interest. All questionnaires developed to address study questions are also clearly stated. The aspects all make perfect sense. The study's results are very informative and indicative to identify prevalence of issues from various dimensions among minority colleague students. Identifying the issue is a critical first step to actions. Such methods can be replicated in other groups to assess severity across demographics. Below are some suggestions from a statistician's perspective:

  1. there are multiple dimensions, such as social stressor, food insecurity, mental and sleep health. It would be helpful to come up with a score using tools such as PCA or structural equation modeling, or at least a multivariable regression to understand each component's contributions.

Response: We have added regression analyses.

  1. COVID is a factor affecting all factors: social stressors, health disparities, food insecurity, etc. Therefore it maybe a multiple layer structure as assumed in the study: COVID->factors->academic performance. On the other hand, academic performance may just another direct consequence from COVID, irrelevant from the assumed factors. There can be causal model attempts to understand the causal pathways.

Response: Given the cross-sectional study a causal model may not be appropriate, however, we have added a regression as recommended.

  1. That's largely in the Hispanic group, may need consistence check among other minority groups.

Response: Thank you for this feedback. We have added to the discussion the importance of addressing the associations in other minority groups as well.

  1. Student's perception is a relatively subjective measure out outcome, compared to GPA, maybe a good idea to repeat the study measurement later. The perception has strong differences between in various comparisons while GPA doesn't show significance among some, which is a reflection of such limitation. Also because student's perception is a relatively new measurement, there's no rules to identify a "scientifically meaningful" change. It may be validated later with more studies.

Response: Thank you for this feedback. We have added to the discussion the importance of addressing future studies. We have also noted in the discussion that GPA is a long-term assessment while term-based performance is more immediate.

  1. A suggestion: stratified has specified statistical meanings. Please try to use "subgroup analysis" if it better describes the analysis performed.

Response: We have updated that language, thank you for this suggestion.

  1. Suggestion: add the size of each category into the table, which will make the table more informative, such as table 3.

Response: We have provided the total sample size and then the following as percent, and thus the size can be inferred as needed; this is the format of reporting consistent with the literature as well as our study approval process.

  1. There's no multiplicity strategy, so the overall type I error rate is not controlled. Also we wish to understand the magnitude of effect, a measurement such as odds ratio, risk difference would help.  

Response: Thank you for the feedback and we have added the odds ratio.

Overall, it is a interesting study investigating a critical topic.

Thank you for your work!

Reviewer 2 Report

It may be worthwhile to explore how minority status, social stressors, and health disparities negatively impact college students' academic performance. However, the submitted manuscript does not present the rationale of the study, the reliability and validity of instruments, research ethics, and results of inferential statistical analysis, so I evaluate that it is difficult to publish in an academic journal. The reasons are described in a little more detail as follows.

1. Looking at the title of the manuscript, it was described that researchers analyzed the negative effects of college students' minority status, social stressors, and health disparities on their academic performance, and illuminated three dimensions of their effects. However, when I read the manuscript, it is not. It seems that the researchers chose that title simply because they included the three predictors in their analysis.

2. You described in detail the impact of the COVID-19 pandemic in the introduction. Is an empirical analysis of that impact included in this study? If not, I think it would be good to deal with it a little in the discussion.

3. The biggest problem is that the introduction does not include any rationale for the logical validity that college students' minority status, social stressors, and health disparities, respectively, will negatively affect their academic performances or achievement.

4. The purpose of the study is not properly presented in the introduction. There are no hypotheses.

5. Sensitive personal information such as GPAs were collected, but there is no mention in the manuscript of obtaining approval from the IRB in relation to research ethics.

6. The instruments that measure the variables are not adequately described, so I am not sure if they are reliable or valid.

7. You cannot achieve the research purpose described in the manuscript title by presenting only very brief descriptive statistics.

Author Response

Dear reviewer,

Thank you for your feedback. We have addressed each and noted our comments in bold for ease of finding.

It may be worthwhile to explore how minority status, social stressors, and health disparities negatively impact college students' academic performance. However, the submitted manuscript does not present the rationale of the study, the reliability and validity of instruments, research ethics, and results of inferential statistical analysis, so I evaluate that it is difficult to publish in an academic journal. The reasons are described in a little more detail as follows.

  1. Looking at the title of the manuscript, it was described that researchers analyzed the negative effects of college students' minority status, social stressors, and health disparities on their academic performance, and illuminated three dimensions of their effects. However, when I read the manuscript, it is not. It seems that the researchers chose that title simply because they included the three predictors in their analysis.

  Response: We have provided in depth discussion of the why we chose the three key variables related to intersectionality.

  1. You described in detail the impact of the COVID-19 pandemic in the introduction. Is an empirical analysis of that impact included in this study? If not, I think it would be good to deal with it a little in the discussion.

 Response: Our discussion has been updated to address additional impact of COVID-19, including long COVID.

  1. The biggest problem is that the introduction does not include any rationale for the logical validity that college students' minority status, social stressors, and health disparities, respectively, will negatively affect their academic performances or achievement.

 Response: We have provided content in the introduction that details the reasons for such variables.

  1. The purpose of the study is not properly presented in the introduction. There are no hypotheses.

 Response: We have provided a purpose and hypothesis statements.

  1. Sensitive personal information such as GPAs were collected, but there is no mention in the manuscript of obtaining approval from the IRB in relation to research ethics.

  Response: The study was approved and such information is provided in required sections of the submission.

  1. The instruments that measure the variables are not adequately described, so I am not sure if they are reliable or valid.

 Responses: Citations for instruments are provided.

  1. You cannot achieve the research purpose described in the manuscript title by presenting only very brief descriptive statistics.

Responses: We have provided bivariate and regression.

Round 2

Reviewer 2 Report

Based on the comments in the first review, I think many things have been revised and supplemented. Overall, I believe that this manuscript can be published in this Journal if the quality of the sentences are improved. Things that need to be supplemented before publication are as follows:

1. You have discuss based only on what you found in this study.

2. Please specify the limitations of your study.

3. The reference style still does not fit this journal. For example, you only need to include the year the journal was published, not the month. And, if the journal has a doi, it must be specified.

Author Response

Thank you for the feedback. Responses are noted below and updates have been made to the manuscript.

Based on the comments in the first review, I think many things have been revised and supplemented. Overall, I believe that this manuscript can be published in this Journal if the quality of the sentences are improved. Things that need to be supplemented before publication are as follows:

  1. You have discuss based only on what you found in this study. Response: We have updated the discussion and added content based on feedback from all reviewers.  

2. Please specify the limitations of your study. Response: Limitations of the study are added. 

3. The reference style still does not fit this journal. For example, you only need to include the year the journal was published, not the month. And, if the journal has a doi, it must be specified. Response: References have been double-checked.